# Shear and hydrostatic stress regulate fetal heart valve remodeling through YAP-mediated mechanotransduction

**Mingkun Wang, Belle Yanyu Lin, Shuofei Sun, Charles Dai, Feifei Long, Jonathan T Butcher***

Meinig School of Biomedical Engineering, Cornell University, Ithaca, United States

**Abstract** Clinically serious congenital heart valve defects arise from improper growth and remodeling of endocardial cushions into leaflets. Genetic mutations have been extensively studied but explain less than 20% of cases. Mechanical forces generated by beating hearts drive valve development, but how these forces collectively determine valve growth and remodeling remains incompletely understood. Here, we decouple the influence of those forces on valve size and shape, and study the role of YAP pathway in determining the size and shape. The low oscillatory shear stress promotes YAP nuclear translocation in valvular endothelial cells (VEC), while the high unidirectional shear stress restricts YAP in cytoplasm. The hydrostatic compressive stress activated YAP in valvular interstitial cells (VIC), whereas the tensile stress deactivated YAP. YAP activation by small molecules promoted VIC proliferation and increased valve size. Whereas YAP inhibition enhanced the expression of cell-cell adhesions in VEC and affected valve shape. Finally, left atrial ligation was performed in chick embryonic hearts to manipulate the shear and hydrostatic stress in vivo. The restricted flow in the left ventricle induced a globular and hypoplastic left atrioventricular (AV) valves with an inhibited YAP expression. By contrast, the right AV valves with sustained YAP expression grew and elongated normally. This study establishes a simple yet elegant mechanobiological system by which transduction of local stresses regulates valve growth and remodeling. This system guides leaflets to grow into proper sizes and shapes with the ventricular development, without the need of a genetically prescribed timing mechanism.

**\*For correspondence:**
jtb47@cornell.edu

**Competing interest:** The authors declare that no competing interests exist.

## Editor's evaluation

This study examines Yap signaling in mouse and chicken embryonic valve development with supporting studies of Yap pathway manipulation and mechanotransduction in valve primordial explants. Calculations of endogenous Yap nuclear/cytoplasmic ratios support conclusions regarding Yap activation status during valve development and under different biomechanical conditions. These studies are novel and provide a clear picture of Yap signaling in embryonic heart valve morphogenesis relative to fluid forces.

## Introduction

Congenital heart disease is the most common birth defects and affects about 0.5–2.0% of the general population (*Tsao et al., 2022*). Congenital heart valve defects accounts for over 25% of all congenital heart disease (*Gilboa et al., 2016*). They can be immediately life threatening at birth or impair the long-term cardiac function in adulthood (*Zimmerman et al., 2020*). Heart valve development starts with endothelial-mesenchymal-transition (EMT), in which valvular endothelial cells (VEC) gain mesenchymal markers and invade into the subendothelial matrix to form endocardial cushions. Post-EMT,

these cells proliferate and differentiate into extracellular matrix (ECM) producing valvular interstitial cells (VIC). With precise regulation of VEC and VIC, the cellularized endocardial cushions undergo ECM remodeling and elongate into thin mature leaflets or cusps. During this process, disturbed growth and remodeling will result in valve malformation and cause clinically relevant cardiac defects (*Wu et al., 2017*; *Gould et al., 2016*; *Lindsey et al., 2015*). Genetic causes of this disturbance have been extensively studied, (*Yu et al., 2019*; *MacGrogan et al., 2018*; *Sifrim et al., 2016*) but can explain less than 20% of clinical cases (*Pierpont et al., 2018*; *Gelb and Chung, 2014*; *Bruneau, 2008*). The importance of mechanical forces in regulating valve development has become well appreciated (*Chow et al., 2022*; *Ahuja et al., 2020*; *Daems et al., 2020*). Oscillatory shear stress (OSS) promotes EMT, and its cellular and molecular mechanisms have been well understood (*O'Donnell and Yutzey, 2020a*). By contrast, the role of mechanical forces in clinically important post-EMT growth and remodeling remain poorly understood.

The flowing blood generates shear and hydrostatic stress on valves. The shear stress is in direct contact with VECs, while the hydrostatic stress causes compression and tension in valves and can be transmitted to VICs. Multiple mechanosensitive signaling pathways have been identified in adult tissues, (*De Belly et al., 2022*; *Souilhol et al., 2020*; *Vining and Mooney, 2017*) but their involvement in valve morphogenesis is unclear. Although it has been shown in zebrafish that shear stress regulates early valvulogenesis (namely EndMT) via bioelectric signaling, (*Fukui et al., 2021*) the near universal VEC in zebrafish heart valves limits study of these later growth and remodeling phases (*O'Donnell and Yutzey, 2020b*). The contributions of VICs and hydrostatic stress, including potential collaborations with VEC, in valvulogenesis are not known. YAP signaling is another widely investigated mechanoactive pathway, (*Dupont et al., 2011*; *Wang et al., 2016*) and transcriptional cofactor of the Hippo pathway (*Moya and Halder, 2019*). YAP signaling is known to regulate cardiac ventricular development by promoting cardiomyocyte proliferation (*Wang et al., 2018*). However, its role in post-EndMT valvular morphogenesis is poorly understood. Moreover, whether YAP pathway responds differently to different forces in different cell types, and how these mechanoresponses collaboratively regulate multiple cell types for a specific tissue morphogenesis, are not known.

Here, we explored the mechanism by which the shear and hydrostatic stress regulate valve growth and remodeling. We used in-vitro and ex-vivo models to decouple the effects of shear and hydrostatic stress on the size and shape of valves. We also studied the role of YAP mediated mechanotransduction in those effects by gain- and loss-of-function tests. To verify our findings in vivo, we performed left atrial ligation (LAL) in chick embryonic hearts. The four-chambered chick hearts develop in a manner that mirrors human heart development. The LAL manipulates mechanical forces in vivo and has been shown to replicate some important features of congenital heart defects (*Gould et al., 2016*; *Salman et al., 2021*; *Ho et al., 2021*). By combining those models, we elaborate how shear and hydrostatic stress regulates VEC and VIC to determine proper size and shape for valves.

## Results

### YAP expression is spatiotemporally regulated

We collected embryonic hearts at different developmental stages from wild type mice and examined the YAP activation in heart valves. We found that YAP was expressed in both mesenchyme and endothelium of outflow tract (OFT) and atrioventricular (AV) cushions at E11.5 (*Figure 1A*). YAP activation in VIC increased significantly at E14.5 (*Figure 1B*) then dropped at E17.5 (*Figure 1C*). This decrease in YAP activation during later remodeling stages was significant in AV valves but insignificant in SL valves, as the development was not uniform across all valves. In VECs on the outflow side, nuclear YAP expression (triangles) increased significantly during later remodeling stages (*Figure 1D*). Although in VECs on the inflow side, cytoplasmic YAP expression (arrows) was stronger throughout all stages.

We also examined the YAP activity in chick embryonic hearts at Hamburger–Hamilton stage (HH) 25, HH30 and HH36 (*Figure 1—figure supplement 1*). It followed the same spatiotemporal pattern. YAP activation in VICs surged at HH30 then dropped at HH36. VECs on the outflow side had an upregulated nuclear YAP expression while YAP expression in VECs on the inflow side was mainly in cytoplasm.

We further examined YAP transcriptional activity to identify the upstream of YAP activation. YAP target genes THBS1, ANKRD1 and PTX3 were elevated by almost 10-fold at E14.5 and HH31 when

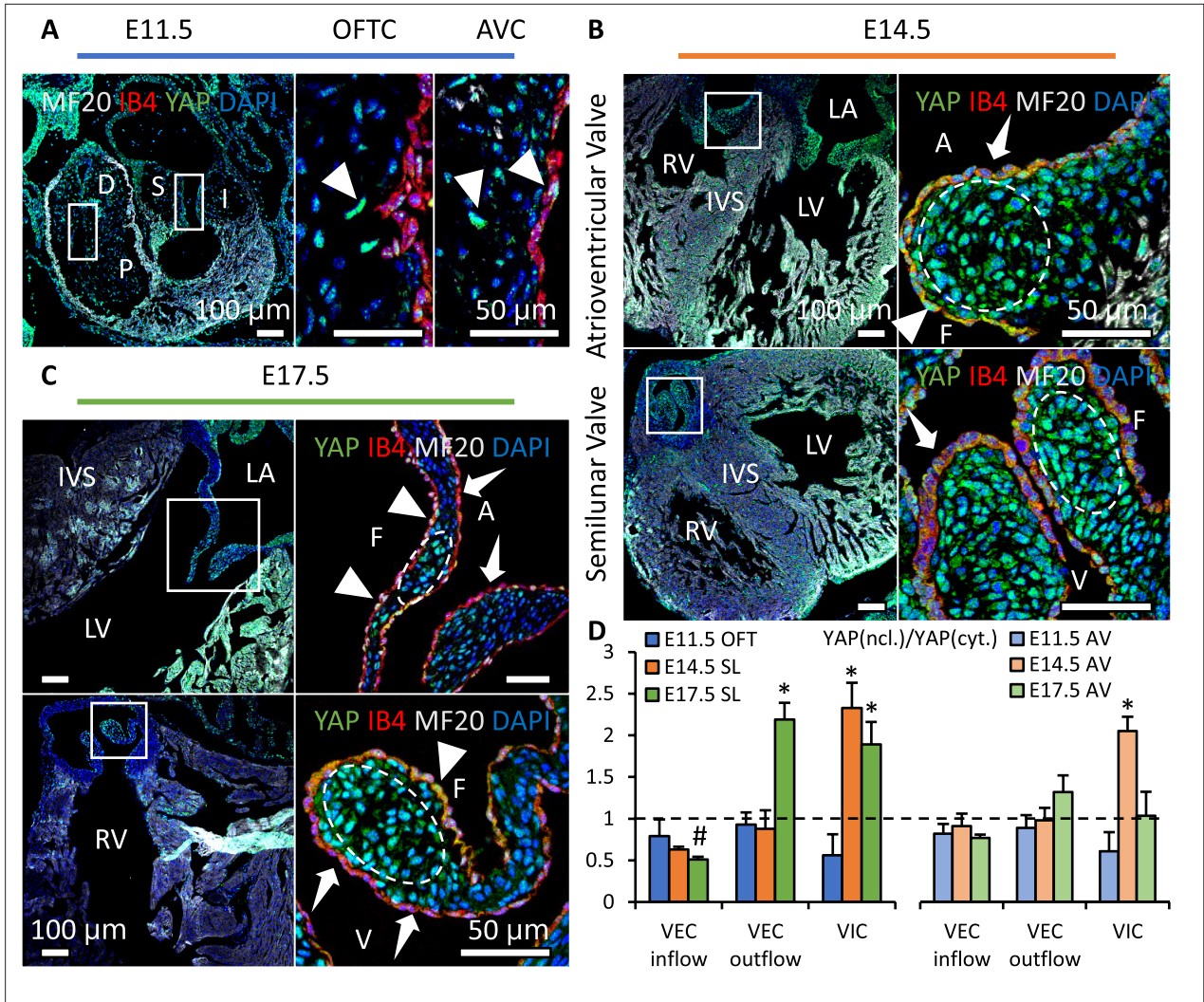

**Figure 1.** YAP expression is spatiotemporally regulated. (**A**). YAP was expressed in both cushion mesenchyme and endothelium (triangles) at E11.5. (**B**). YAP is intensively expressed in VICs (white circles) at E14.5 but seldom detected in VECs (dashed arrows). (**C**). YAP expression remains in VICs on the tip regions and in VECs on the fibrosa side at E17.5 (solid arrows), while disappears in VECs on the ventricularis or ventricularis (dashed arrows) (**D**) Intensity ratios of nuclear vs. cytoplasmic YAP expressions in VECs and VICs of AV and SL valves at different stages. Data are presented by mean ± SEM, n=6 sections from three embryos, *p<0.05, two-way ANOVA tests. OFTC, outflow tract cushion; AVC, atrioventricular cushion; I, inferior cushion; S, superior cushion; D, distal cushion; P, proximal cushion; LA, left atrium; LV, left ventricle; RV, right ventricle; IVS, interventricular septum; V, ventricularis; A, atrialis; F, fibrosa.

The online version of this article includes the following source data and figure supplement(s) for figure 1:

**Source data 1.** YAP activation measurement for *Figure 1D*.

**Figure supplement 1.** YAP expression in chick hearts during embryonic development.

**Figure supplement 2.** qPCR analysis of (**A**) mouse and (**B**).

compared to E11.5 or HH25 cushions, respectively (*Figure 1—figure supplement 2*). The expressions of those genes then reduced significantly during later stages of remodeling. In comparison, gene expressions of LATS1/2, the upstream of YAP in the Hippo pathway, had little change during the valve growth and remodeling. This showed that the YAP activity was largely independent of the Hippo pathway.

## Shear and hydrostatic stress regulate YAP activity

In addition to the co-effector of the Hippo pathway, YAP is also a key mediator in mechanotransduction. Indeed, the spatiotemporal activation of YAP correlated with the changes in the mechanical

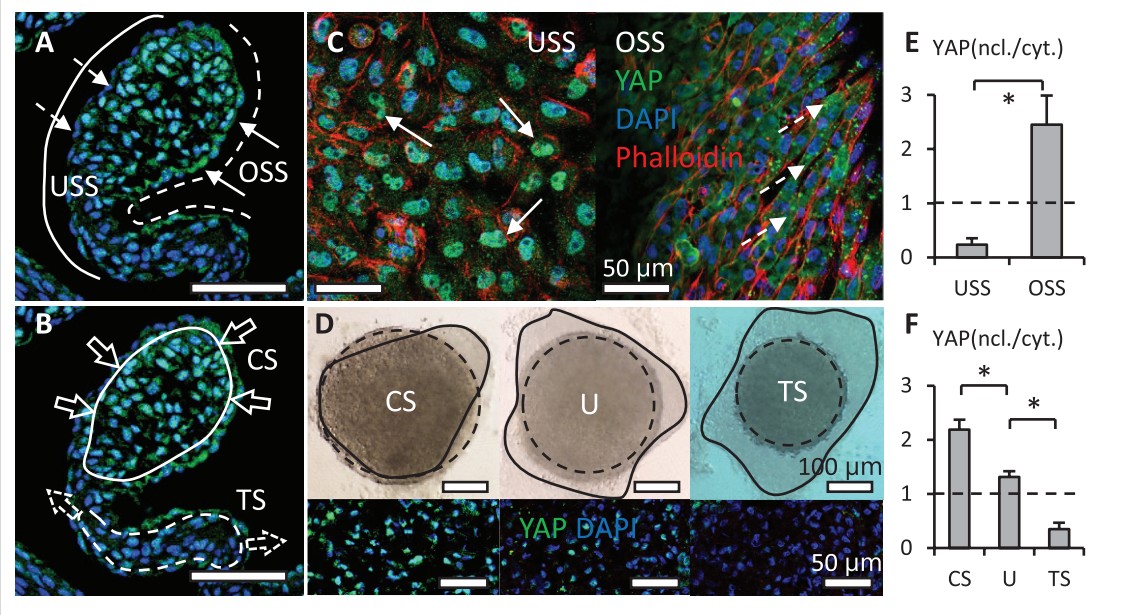

**Figure 2.** Shear stress and hydrostatic pressure regulate YAP activity. (**A**). Unidirectional shear stress (USS) developed on the inflow surface (solid line), where YAP was rarely expressed in the nuclei of VECs (dash arrows). Oscillatory Shear Stress (OSS) developed on the outflow surface (dash line), where VECs with nuclear YAP localized (solid arrows). (**B**). Compressive Stress (CS) was generated in the tips of valves (solid circle), where VICs with nuclear YAP localized. Tensile Stress (TS) is created in the elongated regions (dash circle), where YAP was absent in VIC nuclei. (**C**). When applied on a monolayer of VEC, high USS restricted YAP in cytoplasm, low OSS promoted YAP nuclear localization. (**D**). Cushion explants were cultured under compress stress (CS), tensile stress (TS) and unloaded (U) conditions for 24 hr. Solid outlines describe the explant morphologies at 0 hr, dash outlines describe the explant morphologies at 24 hr. (**E**). Intensity ratios of nuclear vs. cytoplasmic YAP expressions for experiments in (**C, F**). Intensity ratios of nuclear vs. cytoplasmic YAP expressions for experiments in (**D**). Data are presented by mean ± SEM, n=15 explant valves from eight embryos, *p<0.05, two-tailed student t-tests.

The online version of this article includes the following source data for figure 2:

**Source data 1.** YAP activation measurements for *Figure 2E and F*.

environment. During valve remodeling, unidirectional shear stress (USS) developed on the inflow surface of valves, where YAP was rarely expressed in the nuclei of VECs (*Figure 2A*). On the other side, OSS developed on the outflow surface, where VECs with nuclear YAP localized. The YAP activation in VICs also correlated with hydrostatic pressure. The pressure generated compressive stress (CS) in the tips of valves, where VICs with nuclear YAP localized (*Figure 2B*). Although tensile stress (TS) was created in the elongated regions, where YAP was absent in VIC nuclei.

To study the effect of shear stress on the YAP activity in VECs, we applied USS and OSS directly onto a monolayer of freshly isolated VECs. The VEC was obtained from AV cushions of chick embryonic hearts at HH25. The cushions were placed on collagen gels with endocardium adherent to the collagen and incubated to enable the VECs to migrate onto the gel. We then removed the cushions and immediately applied the shear flow to the monolayer for 24 hr. The low stress OSS (2 dyn/cm$^2$) promoted YAP nuclear translocation in VEC (*Figure 2C and E*), while high stress USS (20 dyn/cm$^2$) restrained YAP in cytoplasm.

To study the effect of hydrostatic stress on the YAP activation in VICs, we used media with different osmolarities to mimic the CS and TS. CS was induced by hypertonic condition while TS was created by hypotonic condition, and the Unloaded (U) condition refers to the osmotically balanced media. Notably, in vivo hydrostatic pressure is generated by flowing blood, while in vivo osmotic pressure is generated by cardiac contractility and plays a critical role in the mechanotransduction during valve development (*Vignes et al., 2022*). Despite the different in vivo origination, the osmotic pressure provides a reliable model to mimic the hydrostatic pressure in vitro (*Bassen et al., 2021*). We cultured HH27 AV cushion explants under different loading conditions for 24 hr and found that the trapezoidal cushions adopted a spherical shape (*Figure 2D*). TS loaded cushions significantly compacted, and

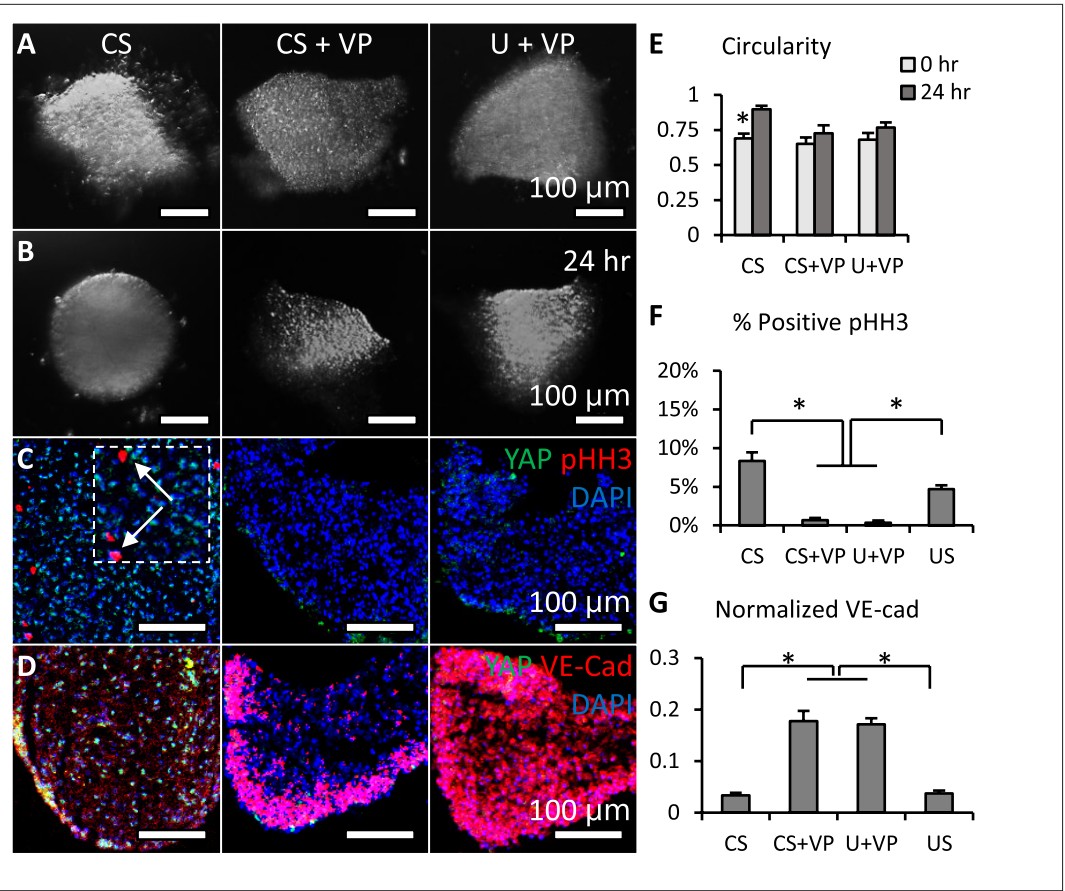

**Figure 3.** Loss of YAP limited cell proliferation and promoted valve shaping. (**A-B**). Cushion explants were cultured under CS (compressive stress), CS +VP (compressive stress +YAP inhibitor), U+VP (unloaded +YAP inhibitor) conditions for 24 hr. (**C**). After 24-hr-culture, explants were stained for YAP (green) and proliferation marker pHH3 (red, arrows), or (**D**). YAP (green) and VE-Cadherin (red). (**E**). Circularity of explants cultured under different stress conditions, which describes how close a valve is to a perfect sphere. (**F**). Percentages of cells expressing pHH3 under different culture conditions. (**G**). Average intensities of VE-Cad expression under different culture conditions, the intensities are normalized to maximum intensity. Data are presented by mean ± SEM, n=15 explant valves from eight embryos, *p<0.05, two-tailed student t-tests.

The online version of this article includes the following source data and figure supplement(s) for figure 3:

**Source data 1.** Data used to generate *Figure 3E, F and G*.

**Figure supplement 1.** Verteporfin and PY-60 treatments.

the YAP activation in VICs of TS-loaded cushions was significantly lower than that in CS loaded VICs (*Figure 2F*).

## Loss of YAP limited cell proliferation and promoted valve shaping

To study the function of YAP in valve growth and remodeling, we added a pharmacological inhibitor of YAP, verteporfin (VP), into the CS (pro-growth) and U conditions. The VP inhibits the interaction between YAP and TEAD, which in turn, blocks transcriptional activation of targets downstream of YAP (*Kagawa et al., 2022*). We cultured cushion explants under CS, CS + VP and U+VP. We confirmed that the concentration of VP we used (5 mM) did not compromise cell viability (*Figure 3—figure supplement 1A*). Successful YAP inhibition (*Figure 3—figure supplement 1C*) reduced the cushion size (*Figure 3A* vs. *Figure 3B*), regardless of the media condition. In contrast to the spherical shape of cushions cultured under CS, valves cultured in media with VP maintained their trapezoidal shape, which was characterized by circularity (*Figure 3E*). Further investigation demonstrated that loss of YAP significantly inhibits the proliferation of VIC, as characterized by pHH3 (*Figure 3C and F*). In addition, the YAP inhibition significantly strengthened the expression of VE-cadherin between VECs

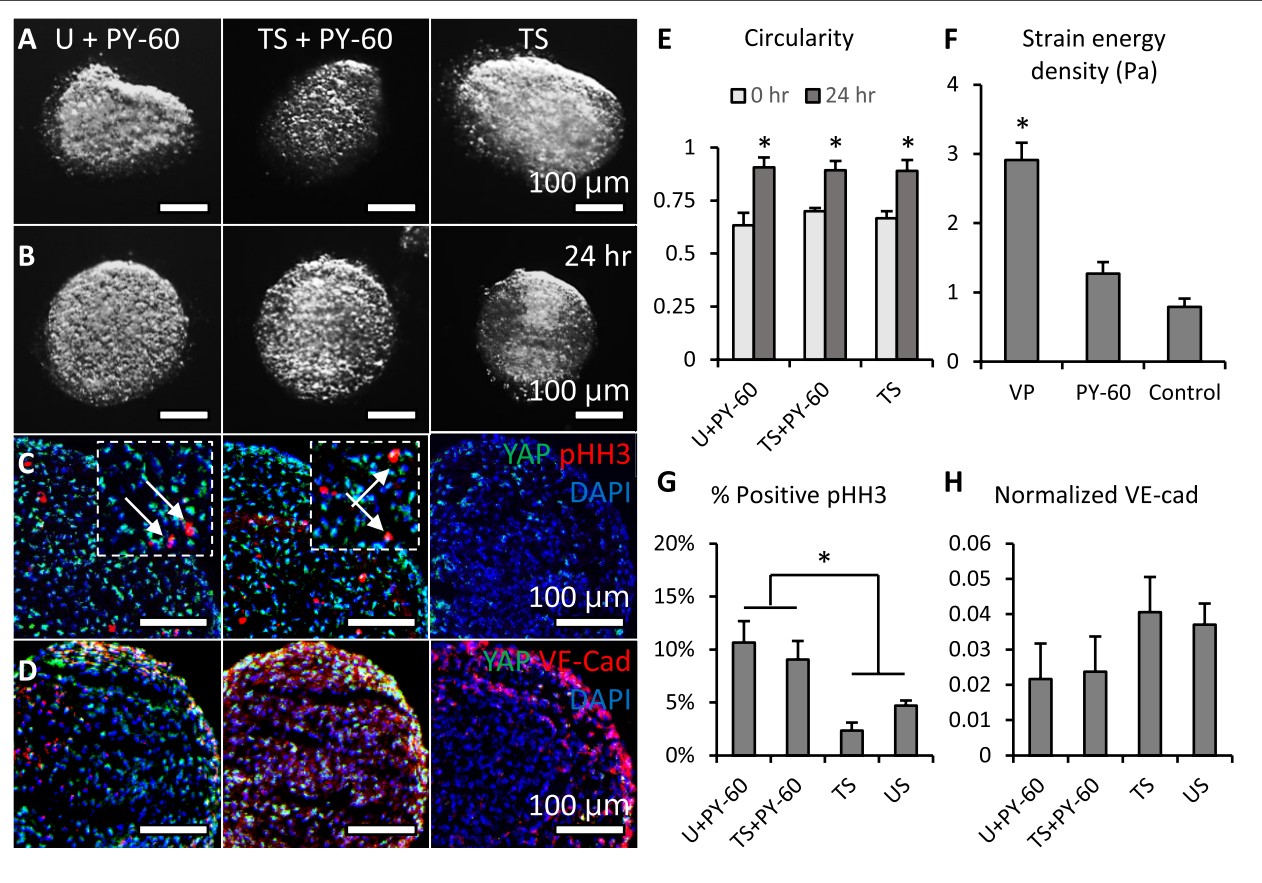

**Figure 4.** Activation of YAP promoted cell proliferation and inhibited valve elongation. (**A-B**). Cushion explants were cultured under U+PY-60 (unloaded +YAP activator), TS +PY-60 (tensile stress +YAP activator), TS (tensile stress) conditions for 24 hr. (**C**). After 24-hr-culture, explants were stained for YAP (green) and proliferation marker pHH3 (red, arrows), or (**D**). YAP (green) and endothelial cell-cell junction VE-Cadherin (red). (**E**). Circularity of explants cultured under different stress conditions, which describes how close a valve is to a perfect sphere. (**F**). Stiffness of cushion explants cultured with YAP activator and inhibitor, which was measured by micropipette aspiration measurement. (**G**). Percentages of cells expressing pHH3 under different culture conditions. (**H**). Average intensities of VE-Cad expression under different culture conditions, the intensities are normalized to maximum intensity. Data are presented by mean ± SEM, n=15 explant valves from eight embryos, *p<0.05, two-tailed student t-tests.

The online version of this article includes the following source data and figure supplement(s) for figure 4:

**Source data 1.** Data used to generate *Figure 4E-H*.

**Figure supplement 1.** Stiffness of cushion explants.

(*Figure 3D and G*). Cushions normally just had a single layer of endothelium, but the YAP inhibited valves showed five or more layers of endothelium.

## Activation of YAP promoted cell proliferation and inhibited valve elongation

For YAP gain-of-function study, a small molecule PY-60 was added into TS (pro-compaction) and U conditions. The PY-60 treatment promoted the association of YAP and TEAD (*Shalhout et al., 2021*). Cell viability did not compromise by PY-60 (10 mM) treatments (*Figure 3—figure supplement 1B*). YAP activation reversed the compaction trend of valves under TS and U conditions and drove a growth in valve size (*Figure 4A* vs. *Figure 4B*). All valves adopted the spherical shape no matter whether they grew or compacted. The pHH3 staining showed that the YAP activation significantly elevated VIC proliferation regardless of loading conditions (arrows, *Figure 4C and G*). The expression of VE-cadherin in the YAP activated endothelium was significantly weaker than that in YAP inhibited endothelium (*Figure 4D*).

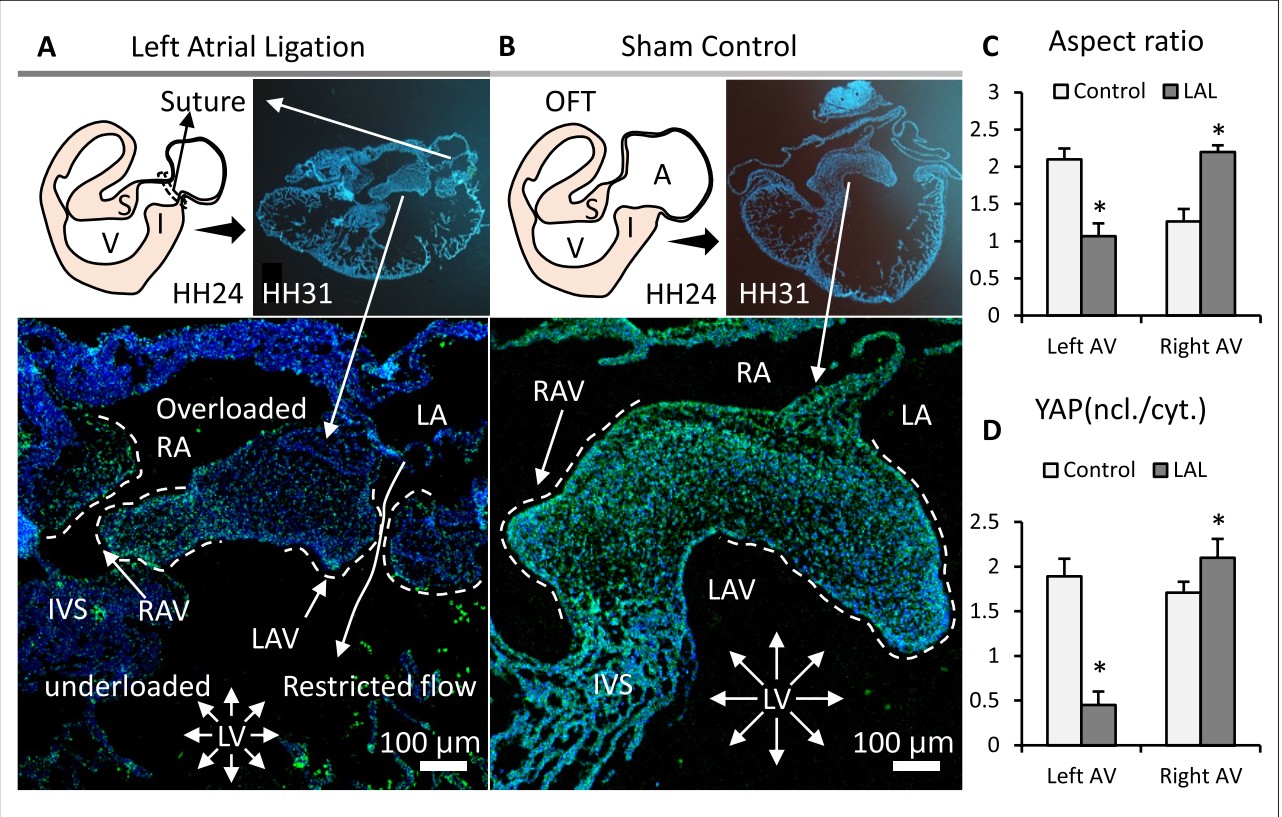

**Figure 5.** Interrupted biomechanics altered YAP activation and led to valve defects in-vivo. (**A**). LAL performed at HH24 led to an underdeveloped left AV septal valve and overdeveloped right AV septal valve at HH31. (**B**). Normally developed sham control hearts. (**C**). Aspect ratios of AV septal valves, which evaluate valve elongation. (**D**). D. Intensity ratios of nuclear vs. cytoplasmic YAP expressions in the septal AV valves of LAL and sham control hearts. Data are presented by mean ± SEM, n=6 sections from three embryos, *p<0.05, two-tail student t-tests. A, atrium; V, ventricle; I, inferior cushion; S, superior cushion; LA, left atrium; RA, right atrium; LV, left ventricle; RV, right ventricle; IVS, interventricular septum; LAV, left atrioventricular septal valve; RAV, right atrioventricular septal valve.

The online version of this article includes the following source data for figure 5:

**Source data 1.** Data used to generate *Figure 5C and D*.

## YAP inhibition promoted an in vivo-like stiffness increase

To assess the role of YAP in valve stiffness, we employed micropipette aspiration to measure the strain energy density of the cushion explants cultured with VP or PY-60 treatment. Micropipette aspiration applies a local vacuum stress and monitors resultant tissue displacement within the tip (*Figure 4—figure supplement 1*). We have previously used this method to measure mechanical properties of chick embryonic valves at different developmental stages (*Buskohl et al., 2012a*). That study showed a nearly linear increase in valve stiffness from HH25 to HH34, when stiffness almost doubled every 24 hr. Here in this study, we found that YAP inhibition also gave a similar stiffness increase during 24-hour-culture (*Figure 4H*). Although both YAP activation and inhibition increased valve stiffness, the stiffness of YAP activated valves was only half of that of YAP inhibited valves.

## In vivo mechanical manipulation led to valve defects and YAP misregulation

To manipulate mechanical forces in vivo, we performed LAL at early stages (HH24) when AV cushions were growing. We collected the hearts during cushion remodeling (HH31) when the valves begin to take shape. The LAL restricted the blood flow in the left ventricle, resulting in a reduced hemodynamic stress and shear stress on left AV valves but augmented forces on the right AV valves. As a result, the LAL valves and hearts (*Figure 5A*) had smaller sizes compared with the control valves and hearts (*Figure 5B*). Specifically, LAL led to an underdeveloped and globular left AV septal valve. While the

right AV septal valve was overdeveloped and elongated. This is opposite to the normal valve development, during which the left AV valve has a much larger size and is more elongated (*Figure 5C*). Unlike the control, where YAP activation was uniformly high in both left and right AV, LAL caused an unbalanced YAP activation: a high activation in right AV while a significantly lower activation in left AV (*Figure 5D*).

## Discussion

Cardiac valves form in response to mechanical forces generated by the flowing blood. These forces include shear and hydrostatic stress. Our results reveal that they can regulate YAP activity in valvular cells, and the mechanically regulated YAP activity can affect the size, shape and stiffness of valves. First, the shear stress regulates YAP activity in the VEC: OSS (oscillatory shear stress) promotes YAP nuclear translocation while USS (unidirectional shear stress) restricts YAP in cytoplasm. The hydrostatic stress regulates YAP activation in the VIC: CS (compressive stress) activated YAP while TS (tensile stress) deactivated YAP. Secondly, these mechanoresponsive YAP activities have morphological functions. YAP activation in VICs promotes their proliferation and increases valve size, YAP deactivation in VICs leads to valve compaction. In terms of shape, cushion explants tend to form a sphere in a way similar to the clustering of cells to minimize the surface tension (*Ninomiya and Winklbauer, 2008*).

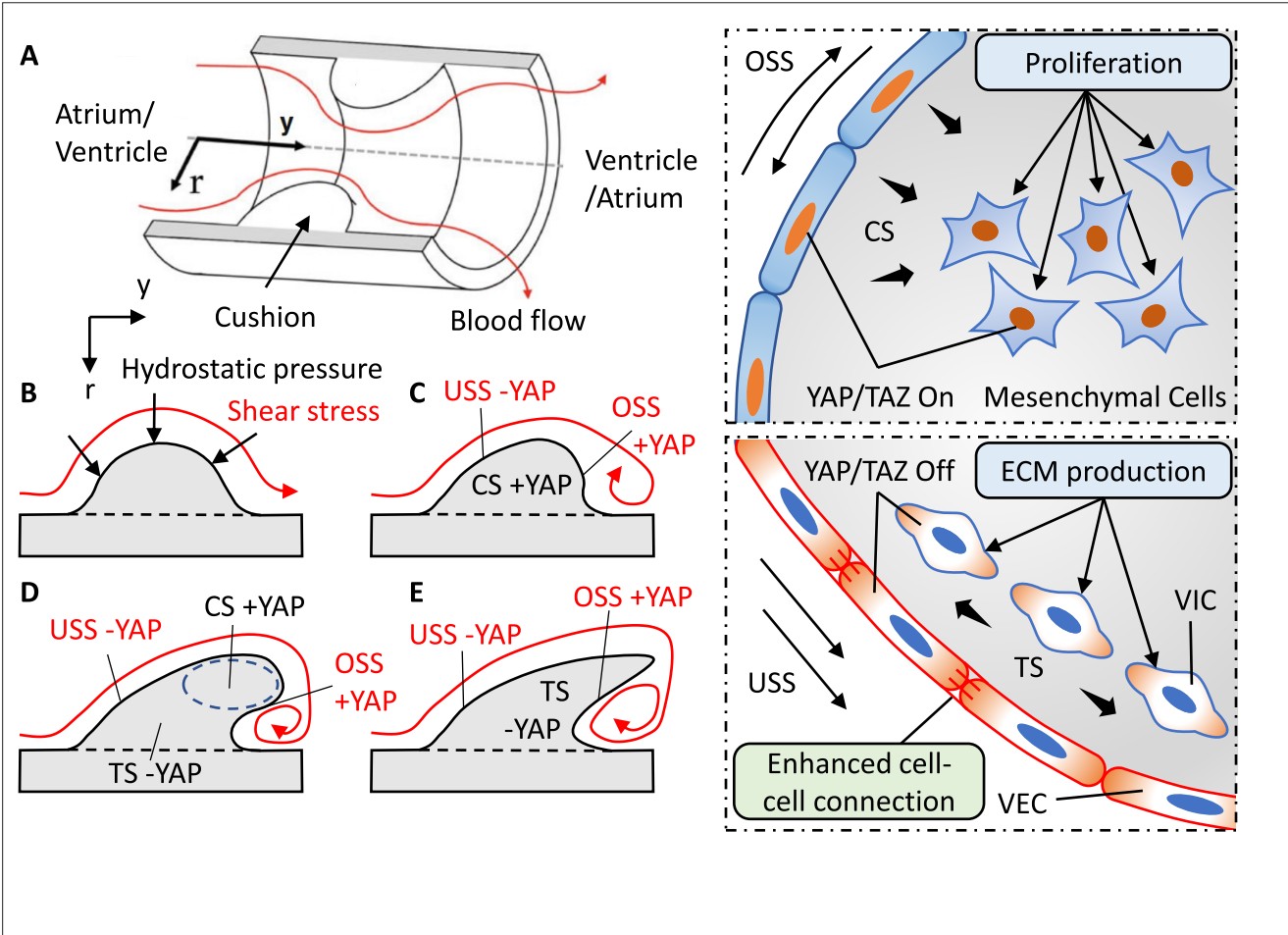

**Figure 6.** Local collaboration of shear and hydrostatic stress can guide complex morphogenesis via YAP signaling. (**A**). Model of a symmetric endocardial cushion in the AV canal or OFT. (**B**). Cross-section of the model shows the flowing blood generates hydrostatic stress and shear stress on cushions. (**C**). Oscillatory Shear Stress (OSS) and Compressive Stress (CS) promoted YAP nuclear translocation in the VEC and VIC, respectively. Unidirectional Shear Stress (USS) inhibited YAP nuclear translocation. CS promoted cushion growth by stimulating the proliferation of mesenchymal cells. (**D**). Shear stress drives cushion remodeling into leaflet structure by modulating the cell-cell adhesions between VECs. (**E**). Tensile Stress (TS) inhibited YAP nuclear translocation worked together with USS for a compact and elongated morphology.

By contrast, YAP inhibition promotes valve shaping and leaves a densely packed endothelium. The densely packed endothelium could be a result of the large shrinkage in surface area, during which the endothelium at 0 hr could be geometrically too large for the valves at 24 hr. Consequently, the endothelium could be forced to cluster into multiple layers. Another possibility is that YAP activity can affect cell-cell junctions between VECs. For example, studies reported that YAP can break cell-cell junctions via SMADs mediated TGF-β signaling during the EMT process (*Zhang et al., 2014*).

In short, shear stress can control valve shape: USS promotes an elongated shape while OSS promotes a globular shape, and hydrostatic stress modulates valve size: CS increases the size while TS leads to compaction. In fact, the simple interactions between mesenchymal and endothelial provide basic elements for tissue morphogenesis (*Hughes et al., 2018*). Here we propose a simple model to achieve the complex leaflet structure via local collaboration between VEC and VIC in response to shear and hydrostatic stress. We use an ideal model (a symmetric cushion in a cylindrical tube) to elaborate this process (*Figure 6A*). The blood flow initiates an asymmetric deformation of the cushion (*Figure 6B*), leaving a gradient stress distribution with highest stress on the top (*Buskohl et al., 2012b*). A combination of OSS and CS drives a globular growth, while USS and CS work together to induce a flat and directional growth (*Figure 6C*). Due to the gradient stress distribution, CS localizes on the top even when the rest part of cushions is not compressed. The localized CS promotes a continuous local growth to fine tune the size and length (*Figure 6D*). USS tightens the inflow surface while OSS loosens the outflow surface. This generates a residual stress in the counterflow direction for the compaction of cushions. When the size and length are adequate, TS collaborates with USS to terminate growth (*Figure 6E*). In this way, the spatiotemporally developed local forces ensure the proper size, length, and thickness of leaflets.

On the other hand, disrupted flow with abnormally developed forces may cause errors in valve growth and remodeling. For example, a combination of delayed OSS and insufficient CS would result in a globular shape and small size, which are the features of underdeveloped valves. Whereas the delayed OSS and overloaded CS could cause a hypertrophic phenotype with globular shape and abnormally big size. In addition, early occurrence of high USS and TS would promote a compacted and elongated shape but with an insufficient length. Such malformations have been verified by in vivo mechanical manipulation. LAL diverted blood flow from the constricted left ventricle toward the untreated right ventricle, decreasing the hemodynamic stress on the left ventricle wall and cushions. This decreased CS in left AV cushions led to insufficient growth. Furthermore, according to the Law of Laplace, the decreased stress also results in a diminished ventricle wall tension that delays the transition from CS to TS, thus hindering cushion compaction. The LAL also altered shear stress environments with a decrease of 40% and over 50% in mean WSS applied on superior cushions and inferior cushions (*Salman et al., 2021*). The level and duration of OSS of the left ventricle also increased significantly with LAL (*Ho et al., 2021*). The delayed emergence of USS could also contribute to the underdeveloped phenotype of the left AV by interfering valve shaping. By contrast, the OSS to USS transition was not significantly affected in the right ventricle, and the right AV cushion compacted and elongated as normal.

In general, YAP works like a mechanobiological switch, converting mechanical signaling into the decision between growth and maturation. When YAP is activated the growth programs are turned on and the maturation programs are suppressed. When YAP is inhibited the growth programs are paused and the maturation programs are released. Unlike the pro-growth function, the pro-maturation side of YAP has been less studied. During valve remodeling, cushions elongate into mature leaflets with increased stiffness. Although both YAP activation and inhibition increased valve stiffness, the stiffness of YAP activated valves was only about half of that of YAP inhibited valves, and only YAP inhibited valves exhibited an in vivo-like stiffness increase. We have previously shown a linear relationship between valve stiffness and valve maturation (*Buskohl et al., 2012a*). This suggests that YAP inhibition promotes a more mature phenotype.

This mechanism explains a major difference between contributions of cell lineage to the EMT and post-EMT growth and remodeling. During the EMT, cell lineage is a determinant factor. Cells derived from the second heart field make the major contribution to the OFT walls, and neural crest cells make a major contribution to the OFT cushions but a minor contribution to AV and intercalated cushions (*Eley et al., 2018*; *Henderson et al., 2020*). However, during the post-EMT growth and remodeling, all valves with various morphologies are formed by identical VECs and VICs. Our study supports that

the collaboration between VECs and VICs, instead of a specific cell lineage, determines the valve growth and remodeling.

In conclusion, our study shows that the spatiotemporally coordinated mechanotransduction of shear and hydrostatic stress is required for proper valve growth and remodeling. The shear and hydrostatic stress can regulate VEC tensions and VIC proliferation by YAP pathways and thus determine the valve size and shape. Malfunctional YAP signaling could cause valve malformation, but improper local mechanical signaling imposes a more important malformation risk, even if the YAP signaling is fully functional and no genes are mutated. The presented mechanobiological system could also open an opportunity to control valve growth and change valve shape by influencing forces or targeting YAP pathway at a specific stage.

## Materials and methods

### Shear stress bioreactor system

The bioreactor included a histology microscope slide, biocompatible double-sided tape (W.W. Grainger), 5-mm-thick silicone sheet (McMaster-Carr) and a sticky-Slide I Luer (Ibidi). Wells with 4 mm diameters were created in the silicone sheet with a 4 mm disposable biopsy punch (Miltex). The magnitude of shear stress is determined by the height of the sticky-Slide I Luer and have been calculated and validated by Ibidi. The sticky-slide I Luer creates a 5 mm x 50 mm channel with various heights. 0.8 mm and 0.4 mm high sticky-slide I Luer were used to create shear stresses of 2 and 20 dyne/cm$^2$, respectively. The components were clamped together with binder clips. The female Luers of the channel slide were connected to the silicone tubing with a 3.2 mm inner diameter (Size 16, Cole-Parmer).

For unidirectional shear stress (USS) experiments, flow was generated using Masterflex L/S Brushless variable-speed digital drive; Masterflex L/S 8-channel, 4-roller cartridge pump head; and Masterflex L/S large cartridges (Cole-Parmer). The peristaltic pump was connected to a pulse dampener (Cole-Parmer) to maintain non-pulsatile unidirectional flow over the samples. The pulse dampener was then connected to the bioreactors. The media was stored in a polycarbonate bottle with a filling/venting cap (Nalge Nunc International) with 80 mL of M199 culture medium, with 1% insulin-selenium-transferrin, Pen/Strep, and 3% chick serum. The cells were exposed to a flow rate of 21.1 mL/min for 24 hr at 37 °C and 5% $CO_2$ within an incubator.

For oscillatory shear stress (OSS) experiments, flow was generated using a NE-1000 syringe pump (New Era Pump Systems, Inc). The bioreactors were connected to a 20 mL syringe (BD Biosciences) that was controlled by the syringe pump. Cells were exposed to shear stress in the forward direction for one-half of a one-second cycle and in the reverse direction for the other half of the cycle at a flow rate of 21.1 mL/min for 24 hr at 37 °C and 5% $CO_2$ within an incubator.

### 3D endocardial cell culture

Collagen gels at a concentration of 2 mg/mL collagen were made using 3 x Dulbecco's Modified Eagle's Medium (Life Technologies), 10% chick serum (Life Technologies), sterile 18 MΩ water, 0.1 M NaOH, and rat tail collagen I (BD Biosciences). An aliquot of the collagen gel solution was pipetted into the wells in the silicone sheet and allowed to solidify for 1 hr at 37 °C and 5% $CO_2$. The dissected outflow tracts were then placed on top of the collagen gel, and excess media was pipetted off to allow for the valve primordia to come in contact with the collagen gel. After 6 hr of incubation at 37 °C and 5% $CO_2$. the valve endocardial cells are repolarized, delaminated, and attached to the surface of the collagen constructs. These endocardial cells were then exposed to USS or OSS at 2 or 20 dyne/cm$^2$ for 24 hr.

### Avian and AV cushion isolation and hanging drop culture system

Atrioventricular cushions (HH25) were dissected from the myocardium of embryonic chick hearts. The explants were cultured in M199 culture medium, 3% chick serum, and 1% insulin-transferrin-selenium. YAP inhibitor verteporfin (10 μg/ml) were purchased from Sigma Aldrich. The explants were placed in 20 μl hanging drops, settled at the apex of the droplets and cultured upside down for 24 hr.

### Left atrial ligation

Fertilized White Leghorn chicken eggs were incubated in a 38 °C forced-draft incubator to Hamburger-Hamilton (HH) stage 21 (3.5 days, Hamburger and Hamilton). The embryo was cultured in an ex-vivo

platform previously described [RG 49]. Briefly, an overhand knot of 10–0 nylon suture loop was placed across a portion of either the right atrium and tightened, partially constricting the left AV orifice. This diverted flow from the constricted inlet toward the untreated inlet, decreasing hemodynamic load on the one side and increasing it on the other [RG 31, 32]. At D7 and D10 (HH31 and HH36, respectively), hearts and/or AVs were fixed, and paraffin sectioned for immunohistochemistry.

## Immunostaining

Chick and mouse hearts and mouse embryos were fixed in 4% paraformaldehyde for overnight, washed with TBS, embedded with paraffin, and sectioned. Sections were deparaffinized and rehydrated. Antigen retrieval was completed using citrate buffer at pH 6.0. Samples were then washed with TBS, permeabilized with 0.3% Triton-X 100 in TBS, and blocked with 3% BSA, 20 mM MgCl, 0.3% Tween 20, 0.3 M Glycine, and 5% Donkey serum in 1xTBS. Samples were then incubated with the primary antibodies at 1:100 dilution with the blocking solution overnight. Primary antibodies used include YAP (mouse DSHB YAP1 8J19, mouse Santa Cruz sc-101199, rabbit cell signaling #14074), Lef1 (rabbit Cell Signaling #2230), pHH3 (rabbit Cell Signaling #9701), VE-cad (abcam ab33168), MF-20 (mouse DSHB MF 20), IB4 (Vector B-1205-.5), Phalloidin (Invitrogen A12379). Samples were washed with 0.3% Triton-X 100 in TBS and incubated with secondary antibodies at 1:100 dilution with 5% BSA in TBS at room temperature for 1 hr. Secondary antibodies used include species-specific Alexa Fluor 568 or 647. Samples were then washed again for 3x10 min with TBS and stained with DAPI. Images were taken with Zeiss LSM 710 confocal microscope.

## Micropipette aspiration experiments

Mechanical properties of cushion explants were measured by micropipette aspiration. A glass micropipette (rp ≥35 µm) was placed adjacent to the cushion surface collinear with the AV canal axis. Vacuum stress was incrementally applied via a 200 µL pipette calibrated with a custom manometer. Previous strain history was mitigated by preconditioning with 20 cycles of low pressurization (<1 Pa). The preconditioning step ensured the tissue and pipette tip were in full contact. Incremental stress loads were then applied, at which images were captured for each static load at ×150 magnification using a Zeiss Discovery v20 stereo microscope. The aspirated length was measured using calibrated images in NIH ImageJ. An experimental 'stretch ratio', $\lambda = (L+r_p)/r_p$ was defined by normalizing the aspirated length to the pipette radius. The experiment stretch ratio is a measure of geometry change during aspiration, which is related, but not identical to the local stretch of the tissue. The ΔP versus $\lambda$ curves were presented.

## Real-time PCR

RNA extractions were performed using a Qiagen total RNA purification kit (Qiagen, Valencia, CA) and RNA was reverse transcribed to cDNA using the SuperScript III RT-PCR kit with oligo(dT) primer (Invitrogen). Sufficient quality RNA was determined by an absorbance ratio A260/A280 of 1.8–2.1, while the quantity of RNA was determined by measuring the absorbance at 260 nm (A260). Real-time PCR experiments were conducted using the SYBR Green PCR system (Biorad, Hercules, CA) on a Biorad CFX96 cycler, with 40 cycles per sample. Cycling temperatures were as follows: denaturing, 95 C; annealing, 60 C; and extension, 70 C. Expression of mouse genes are normalized to GAPDH, and chick genes are normalized to 18 S. Sequences of primers are included below.

| Gene | Forward primer (5' - 3') | Reverse primer (5' - 3') | Accession No. |
|---|---|---|---|
| Mouse GAPDH | CTCCTGCACCACCAACTGCT | GGGCCATCCACAGTCTTCTG | NM_001289726.1 |
| Mouse ANKRD1 | AGTAGAGGAACTGGTCACTG | TGGGCTAGAAGTGTCTTCAGAT | NM_013468 |
| Mouse LATS1 | CTCTGCACTGGCTTCAGATG | TCCGCTCTAATGGCTTCAGT | NM_010690.1 |
| Mouse LATS2 | ACATTCACTGGTGGGGACTC | GTGGGAGTAGGTGCCAAAAA | NM_015771.2 |
| Mouse PTX3 | CGAAATAGACAATGGACTTCATCC | CATCTGCGAGTTCTCCAGCATG | NM_002852 |
| Mouse LATS1 | CTCTGCACTGGCTTCAGATG | TCCGCTCTAATGGCTTCAGT | NM_010690.1 |
| Mouse LATS2 | ACATTCACTGGTGGGGACTC | GTGGGAGTAGGTGCCAAAAA | NM_015771.2 |

*Continued on next page*

*Continued*

| Gene | Forward primer (5′ - 3′) | Reverse primer (5′ - 3′) | Accession No. |
|---|---|---|---|
| Mouse THBS1 | GGTAGCTGGAAATGTGGTGCGT | GCACCGATGTTCTCCGTTGTGA | NM_011580.4 |
| Chicken 18 S | TAGTTGGTGGAGCGATTTGTCT | CGGACATCTAAGGGCATCACA | AF173612.1 |
| Chicken ANKRD1 | CCTTCCCACAGCTCTCAATAG | GATAAAGGGCTCATGGACAGAG | NM_204405.2 |
| Chicken LATS1 | GTTCTGCCAACAGCAAGTTTAG | GCTGGTGTGACTCTGTCTATTT | XM_004935606.5 |
| Chicken LATS2 | TCTTCCAACAGCAAGCACAC | AAGCTCCAGTCTGATCCACC | XM_015279299.2 |
| Chicken PTX3 | CTGAGACACTCGGAGCATTTAT | CAATCCCTATGAGATCCAGCTG | NM_001017413.1 |
| Chicken THBS1 | CCACCTTCAGGAGTGTGATAAG | CCGCAAAGCAGGGATTAGA | NM_001199453.2 |

## Statistics

Images were analyzed using ImageJ. Results are presented as mean ± SD and compared using either an ANOVA tests with Tukey post hoc paired or two-tailed student t-tests. Differences were considered significant at $p \leq 0.05$. For immunofluorescence staining, heart sections from $n \geq 5$ embryonic hearts were used for quantification. For cushion explants culture, $n \geq 5$ independent cultures per treatment condition and four to six dozen chick embryos pooled for each experiment. For shear flow experiments, $n \geq 5$ endocardial patches per shear flow condition. For LAL surgery, $n \geq 3$ survival embryos for LAL control and sham control.

## Acknowledgements

This work was supported by NIH (grants HL128745, HL143247, HL160028 to JTB), NSF URoL (to JTB), Additional Ventures Single Ventricle Research Fund (to JTB), American Heart Association (grant 821615 to MW) and by biotechnology center via NYSTEM C029155 and NIH S10OD018516.

## Additional information

### Funding

| Funder | Grant reference number | Author |
|---|---|---|
| National Institutes of Health | HL128745 | Jonathan T Butcher |
| National Institutes of Health | HL143247 | Jonathan T Butcher |
| National Institutes of Health | HL160028 | Jonathan T Butcher |
| American Heart Association | 821615 | Mingkun Wang |
| National Science Foundation | URoL | Jonathan T Butcher |
| Additional Ventures | Single Ventricle Research Fund | Jonathan T Butcher |

The funders had no role in study design, data collection and interpretation, or the decision to submit the work for publication.

### Author contributions

Mingkun Wang, Conceptualization, Data curation, Formal analysis, Investigation, Visualization, Methodology, Writing - original draft, Writing - review and editing; Belle Yanyu Lin, Formal analysis, Validation, Investigation; Shuofei Sun, Charles Dai, Feifei Long, Investigation; Jonathan T Butcher, Conceptualization, Resources, Supervision, Funding acquisition, Writing - review and editing

## Author ORCIDs

Mingkun Wang http://orcid.org/0000-0002-9273-7645
Jonathan T Butcher http://orcid.org/0000-0002-9309-6296

## Decision letter and Author response

Decision letter https://doi.org/10.7554/eLife.83209.sa1
Author response https://doi.org/10.7554/eLife.83209.sa2

## Additional files

### Supplementary files
• MDAR checklist

### Data availability

All antibodies, chemicals, and sequences of primers in the study are listed in the Methods. Figure 1—source data 1 and Figure 2—source data 1 Figure 3—source data 1, Figure 4—source data 1 and Figure 5—source data 1 contain the numerical data used to generate the figures.

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
