## [Editor Report]

This study examines Yap signaling in mouse and chicken embryonic valve development with supporting studies of Yap pathway manipulation and mechanotransduction in valve primordial explants. Calculations of endogenous Yap nuclear/cytoplasmic ratios support conclusions regarding Yap activation status during valve development and under different biomechanical conditions. These studies are novel and provide a clear picture of Yap signaling in embryonic heart valve morphogenesis relative to fluid forces.

---

## [Decision Letter]

**Decision letter after peer review:**

Thank you for submitting your article "Shear and hydrostatic stress regulate fetal heart valve remodeling through YAP-mediated mechanotransduction" for consideration by *eLife*. Your article has been reviewed by 3 peer reviewers, one of whom is a member of our Board of Reviewing Editors, and the evaluation has been overseen by Didier Stainier as the Senior Editor. The following individual involved in the review of your submission has agreed to reveal their identity: Julien Vermot (Reviewer #3).

Essential revisions:

1) Improved analysis of Yap activation status relative to valve areas under different mechanical stresses is needed. Immunostaining for total Yap is inadequate to support the major findings in the study.

2) Additional manipulation of Yap activation is needed to support the major conclusion that Yap-medicated mechanotransduction has a role in fetal heart valve remodeling response to shear and hydrostatic stress.

3) Additional major conclusions in the study are not supported by rigorous data. Please see the reviewers' comments.

*Reviewer #1 (Recommendations for the authors):*

1. Additional data are needed to determine Yap activation status as might be indicated by nuclear/cytoplasmic ratios and Yap phosphorylation. The Yap immunostaining is widespread and does not seem to relate to localized mechanical forces. Other indicators of Hippo signaling status might be informative.

2. There are no clear conclusions regarding Yap signaling status, cell proliferation, or valve morphogenesis under different biomechanical conditions. This is described in Figures 3A, B, but localized Yap activation in specific regions or cell types is not clear in these panels. The data are confusing largely due to the widespread Yap expression shown in VIC, VEC, and other cell types throughout the valves.

3. Conclusions regarding endothelial folding and altered junctions between endothelial cells are not supported by data.

4. It is not clear what tissue rounding or increased "stiffness" of valve explants means in the context of leaflet elongation, ECM remodeling, cell density, or growth.

5. In the chicken embryo heart unloading study (Figure 6), Yap is widely expressed in the controls, but regions of Yap activation are not apparent. It is not clear how the ratios of positive Yap were calculated or if there were specific regions of activation in response to specific blood flow dynamics.

6. Figure 7 is described as a "pseudo model" but it should probably be just a "model". The dynamics of Yap activation shown in the model were not clearly demonstrated in the figures.

*Reviewer #2 (Recommendations for the authors):*

Addressing the weaknesses outlined below would greatly strengthen the significance and impact of the work.

Identified weaknesses of the manuscript include:

– Figure 1 demonstrates dynamic YAP expression in the developing mouse; however, all experimental studies are performed in chick. Comparative temporal and spatial expression studies in the avian model should be performed

– The relevance of YAP1 and LEF1 co-staining (or lack of) in Figure 2 adds little value to the study. At E11.5, yellow cells are identified suggesting that there is an overlap. The authors comment that the data supports "that YAP has different functions at early EndMT than later remodeling stages". However, based on this expression data, this conclusion is difficult to interpret and support.

– In Figure 3, data presented in panel D should be quantitated. In addition, what are the axis referring to on the graphs of Figures 3E, F?

– In the text referring to Figure 3, the authors discuss examining the "YAP signaling pathway" but this is not being examined here. In addition, the authors claim that USS restrains YAP in the cytoplasm, but this is not detectable.

Throughout the study, the authors solely rely on the quantitation of nuclear YAP expression based on IHC. An addition supporting experimental readout is highly recommended, including nuclear/cytoplasmic extraction and western blot to detect YAP in cellular fractions for Figures 3C, and 3D.

– For Figures 4E, 4F, 4G, 5E-H, the US control is needed.

– In Figure 4B, there is a large decrease in circularity with YAP inhibition, yet proliferation changes appear minimal. What other YAP-dependent mechanisms might contribute to this drastic change in valve morphology? Likewise, Figure 4F indicates 10% pHH3 reactivity, yet this is not represented in Figure 4C. Which cells are proliferating?

– The phenotype in Figure 4D is interesting and unusual. The authors conclude that loss of YAP leads to endothelium folding, but again, there is little evidence to support this. If this is true, then additional supporting readouts are needed including another endothelial cell marker. How do the authors explain an increase in VE-cadherin with so little pHH3 reactivity in the VECs?

– On Page 4, the authors comment that VP treatment led to "successful YAP inhibition" but this data is not included.

– When referring to Figure 5, the authors comment that PY-60 treatment dose-dependently promoted the association of YAP and TEAD. This data is not included.

– Figure 5E is confusing and suggests that YAP gain of function has no effect on stress-induced increases in circularity, yet the authors conclude that YAP activation reverses the compaction of valves under TS and U conditions.

– The conclusion of findings from Figure 5D is that "VE-cadherin in the YAP activated endothelium was significantly weaker…showing that YAP activation led to a relaxed endothelium." Again, this conclusion needs to be supported by additional data.

– Figure 5H is mislabeled in the text.

– The authors should comment further on how and why YAP inhibition decreases the strain energy density. How can YAP influence the material properties of avian valves?

---

## [Author Response]

Reviewer #1 (Recommendations for the authors):1. Additional data are needed to determine Yap activation status as might be indicated by nuclear/cytoplasmic ratios and Yap phosphorylation. The Yap immunostaining is widespread and does not seem to relate to localized mechanical forces. Other indicators of Hippo signaling status might be informative.

We thank the reviewer for the constructive suggestion. We have now analyzed the intensity ratios of YAP expressions in nuclei versus cytoplasm as suggested to quantify YAP activation status and revised Figure 1, Figure 2 (previous Figure 3), and Figure 5 (previous Figure 6).

We have also added qPCR data of YAP target genes THBS1, ANKRD1 and PTX3, as well as YAP upstream in Hippo cascade LATS1 and LATS2, into a new supplementary Figure 1—figure supplementary 2. Expressions of THBS1, ANKRD1 and PTX3 in mouse cushions markedly upregulated from E11.5 to E14.5 but fell sharply from E14.5 to E17.5. By contrast, LATS1 and LATS2 did not change statistically. Fold changes of those genes in chick cushions followed the same trend. These data were also consistent with YAP activation measured from immunofluorescence. The results showed that YAP was independent of the Hippo pathway. It is worth noting that the PCR data mainly reflect gene transcriptions of the VIC, as it was the dominant cell population in valves.

We have also revised the first two sections of Results:

“YAP expression is spatiotemporally regulated

We collected embryonic hearts at different developmental stages from wild type mice and examined the YAP activation in heart valves. We found that YAP was expressed in both mesenchyme and endothelium of outflow tract (OFT) and atrioventricular (AV) cushions at E11.5 (Figure 1A). YAP activation in VIC increased significantly at E14.5 (Figure 1B) then dropped at E17.5 (Figure 1C). This decrease in YAP activation during later remodeling stages was significant in AV valves but insignificant in SL valves, as the development was not uniform across all valves. In VECs on the outflow side, nuclear YAP expression (triangles) increased significantly during later remodeling stages (Figure 1D). Whereas in VECs on the inflow side, cytoplasmic YAP expression (arrows) was stronger throughout all stages.

We also examined the YAP activity in chick embryonic hearts at Hamburger–Hamilton stage (HH) 25, HH30 and HH36 (Figure 1—figure supplementary 1). It followed the same spatiotemporal pattern. YAP activation in VICs at HH30 then dropped at HH36. VECs on the outflow side had an upregulated nuclear YAP expression with development while YAP expression in VECs on the inflow side was mainly in cytoplasm.

We further examined YAP transcriptional activity to identify the upstream of YAP activation. YAP target genes THBS1, ANKRD1 and PTX3 were elevated by almost 10-fold at E14.5 and HH31 when compared to E11.5 or HH25 cushions, respectively (Figure 1—figure supplementary 2). The expressions of those genes then reduced significantly during later stages of remodeling. In comparison, gene expressions of LATS1/2, the upstream of YAP in the Hippo pathway, had little change during the valve growth and remodeling. This result supports that YAP activation is operated independently of the Hippo pathway.”

2. There are no clear conclusions regarding Yap signaling status, cell proliferation, or valve morphogenesis under different biomechanical conditions. This is described in Figures 3A, B, but localized Yap activation in specific regions or cell types is not clear in these panels. The data are confusing largely due to the widespread Yap expression shown in VIC, VEC, and other cell types throughout the valves.

We used the dash lines for clarity of surface flow domains, but these may have covered YAP expression in the VEC. We have now improved the annotations in Figure 2A and 2B. In Figure 2A, solid lines are used for the USS domain and dash lines for the OSS domain. Solid arrows highlight nuclear YAP, and dash arrows highlight nuclei without YAP. In Figure 2B, the CS domain is highlighted by a solid circle with force direction and the TS domain by a dash circle with force direction. We have also changed the color scheme in Figure 2 for a better visualization.

3. Conclusions regarding endothelial folding and altered junctions between endothelial cells are not supported by data.

We appreciate the reviewer for pointing out our lack of clarity. The term “endothelial folding” was poorly chosen. In fact, by “folding”, we meant the shrinkage of surface areas. Due to the compaction of cushion explants, their surface areas became much

smaller, and the shrinkage is approximately proportional to (*d/D*)^2^. Even a small compaction will cause a large reduction in surface areas. As a result, the endothelium at 0-hour was geometrically too large for the cushions at 24-hour. Therefore, the endothelium could be forced to cluster into multiple layers. Indeed, we observed extra thick and densely packed VE-cadherin positive layers in YAP inhibited valves. Another possibility is that YAP activity can affect cell-cell junctions between VECs. For example, some studies reported that YAP can break cell-cell junctions between VECs via SMADs mediated TGF-β signaling during the EMT process. The detailed mechanisms by which YAP affects the cell-cell junctions and endothelium behaviors is important but requires further studies to elaborate.

We have moved this discussion from the first section of the Results to the Discussion, we have also revised titles of the third and fourth sections of the Results and titles of Figure 3 and Figure 4.

The revised third section of the Results, as well as the title of Figure 3 are as follows:

“Loss of YAP limited cell proliferation and promoted valve shaping

To study the function of YAP in valve growth and remodeling, we added a pharmacological inhibitor of YAP, verteporfin (VP), into the CS (pro-growth) and U conditions. The VP inhibits the interaction between YAP and TEAD, which in turn, blocks transcriptional activation of targets downstream of YAP. (31) We cultured cushion explants under CS, CS+VP and U+VP. We confirmed that the concentration of VP we used (5mM) does not influence cell viability (Figure 3—figure supplementary 1A). Successful YAP inhibition (Figure 3—figure supplementary 1C) reduced the cushion size (Figure 3A vs. Figure 3B), regardless of the media condition. In contrast to the spherical shape of cushions cultured under CS, cushions cultured in media with VP maintained their trapezoidal shape, which was characterized by circularity (Figure 3E). Further investigation demonstrated that loss of YAP significantly reduces the expression of the proliferation maker, pHH3 in the VIC (Figure 3C, 3F). In addition, the YAP inhibition significantly strengthened the expression of VE-cadherin between VECs (Figure 3D, 3G). Cushions normally just had a single layer of endothelium, but the YAP inhibited valves showed five or more layers of endothelium."

The revised title of fourth section of the Results and Figure 4 is:

“Activation of YAP promoted cell proliferation and inhibited valve elongation”

The new first paragraph of the Discussion is as follows:

“Cardiac valves form in response to mechanical forces generated by the flowing blood. These forces include shear and hydrostatic stress. Our results reveal that they can regulate YAP activity in valvular cells, and the mechanically regulated YAP activity can affect the size, shape and stiffness of valves. First, the shear stress regulates YAP activity in the VEC: OSS (oscillatory shear stress) promotes YAP nuclear translocation while USS (unidirectional shear stress) restricts YAP in cytoplasm. The hydrostatic stress regulates YAP activation in the VIC: CS (compressive stress) activated YAP while TS (tensile stress) deactivated YAP. Secondly, these mechanoresponsive YAP activities have morphological functions. YAP activation in VICs promotes their proliferation and increases valve size, YAP deactivation in VICs leads to valve compaction. In terms of shape, cushion explants tend to form a sphere in a way similar to the clustering of cells to minimize the surface tension (32). By contrast, YAP inhibition promotes valve shaping and leaves a densely packed endothelium. The densely packed endothelium could be a result of the large shrinkage in surface area, during which the endothelium at 0-hour could be geometrically too large for the valves at 24-hour. Consequently, the endothelium could be forced to cluster into multiple layers. Another possibility is that YAP activity can affect cell-cell junctions between VECs. For example, studies reported that YAP can break cell-cell junctions via SMADs mediated TGF-β signaling during the EMT process (33).”

4. It is not clear what tissue rounding or increased "stiffness" of valve explants means in the context of leaflet elongation, ECM remodeling, cell density, or growth.

During valve remodeling, cushions elongate into mature leaflets with increased stiffness. We have previously shown a nearly linear relationship between strain energy and leaflet elongation in-vivo.

The new fourth paragraph of Discussion is as follows:

“In general, YAP works like a mechanobiological switch, converting mechanical signaling into the decision between growth and maturation. When YAP is activated the growth programs are turned on and the maturation programs are suppressed. When YAP is inhibited the growth programs are paused and the maturation programs are released. Unlike the pro-growth function, the pro-maturation side of YAP has been less studied. During valve remodeling, cushions elongate into mature leaflets with increased stiffness. Although both YAP activation and inhibition increased valve stiffness, the stiffness of YAP activated valves was only about half of that of YAP inhibited valves, and only YAP inhibited valves exhibited an in-vivo-like stiffness increase. We have previously shown a linear relationship between valve stiffness and valve maturation (32). This suggests that YAP inhibition promotes a more mature phenotype.”

5. In the chicken embryo heart unloading study (Figure 6), Yap is widely expressed in the controls, but regions of Yap activation are not apparent. It is not clear how the ratios of positive Yap were calculated or if there were specific regions of activation in response to specific blood flow dynamics.

The control was at HH31, when the YAP activation surged to the highest level. This was consistent with the qPCR data that the fold change of YAP target gene expressions was highest at E14.5 or HH31. We have now used the Intensity ratios of nuclear to cytoplasmic YAP to clarify this confusion.

6. Figure 7 is described as a "pseudo model" but it should probably be just a "model". The dynamics of Yap activation shown in the model were not clearly demonstrated in the figures.

We thank the reviewer for this reminder. We have now removed the word “pseudo”. we have also revised Figure 6B-6E (previously Figure 7) to reflect the dynamics of YAP activity during remodeling.

Reviewer #2 (Recommendations for the authors):Addressing the weaknesses outlined below would greatly strengthen the significance and impact of the work.Identified weaknesses of the manuscript include:– Figure 1 demonstrates dynamic YAP expression in the developing mouse; however, all experimental studies are performed in chick. Comparative temporal and spatial expression studies in the avian model should be performed

We apologize if these requested experiments and results weren’t suitably called out in the manuscript. Immunostainings of chick embryonic heart valves at different developmental stages were given in Figure S4, which has now been moved to Figure 1—figure supplementary 1. We have also revised the first section of Results to present the data in Figure 1—figure supplementary 1 right after Figure 1:

“YAP expression is spatiotemporally regulated

We collected embryonic hearts at different developmental stages from wild type mice and examined the YAP activation in heart valves. We found that YAP was expressed in both mesenchyme and endothelium of outflow tract (OFT) and atrioventricular (AV) cushions at E11.5 (Figure 1A). YAP activation in VIC increased significantly at E14.5 (Figure 1B) then dropped at E17.5 (Figure 1C). This decrease in YAP activation during later remodeling stages was significant in AV valves but insignificant in SL valves, as the development was not uniform across all valves. In VECs on the outflow side, nuclear YAP expression (triangles) increased significantly during later remodeling stages (Figure 1D). Whereas in VECs on the inflow side, cytoplasmic YAP expression (arrows) was stronger throughout all stages.

We also examined the YAP activity in chick embryonic hearts at Hamburger–Hamilton stage (HH) 25, HH30 and HH36 (Figure 1—figure supplementary 1). It followed the same spatiotemporal pattern. YAP activation in VICs at HH30 then dropped at HH36. VECs on the outflow side had an upregulated nuclear YAP expression with development while YAP expression in VECs on the inflow side was mainly in cytoplasm.

We further examined YAP transcriptional activity to identify the upstream of YAP activation. YAP target genes THBS1, ANKRD1 and PTX3 were elevated by almost 10-fold at E14.5 and HH31 when compared to E11.5 or HH25 cushions, respectively (Figure 1—figure supplementary 2). The expressions of those genes then reduced significantly during later stages of remodeling. In comparison, gene expressions of LATS1/2, the upstream of YAP in the Hippo pathway, had little change during the valve growth and remodeling. This result supports that YAP activation is operated independently of the Hippo pathway.”

– The relevance of YAP1 and LEF1 co-staining (or lack of) in Figure 2 adds little value to the study. At E11.5, yellow cells are identified suggesting that there is an overlap. The authors comment that the data supports "that YAP has different functions at early EndMT than later remodeling stages". However, based on this expression data, this conclusion is difficult to interpret and support.

We understand the reviewer’s comments regarding LEF1, and we have removed the related sections in Results as requested.

– In Figure 3, data presented in panel D should be quantitated. In addition, what are the axis referring to on the graphs of Figures 3E, F?

we apologize if this information wasn’t suitably called out in the manuscript. We have revised the Yap analysis as requested by Reviewer 1. Figure 3E is the quantification of YAP activation in the monolayer VEC experiments (Figure 3C). Figure 3F is the quantification of YAP activation in the ex-vivo valve explant experiments (Figure 3D). The area ratios that describe the valve compaction is not included here, as we have published detailed compaction data in previous studies:

Bassen D, Wang M, Pham D, Sun S, Rao R, Singh R, et al. Hydrostatic mechanical stress regulates growth and maturation of the atrioventricular valve. Development. 2021;148(13).

Therefore, in this study, compaction levels under different stress conditions were given directly. We have also revised the Figure 3 (now Figure 2), the axis of E and F refers to the intensity ratio of nuclear vs. cytoplasmic YAP expressions.

– In the text referring to Figure 3, the authors discuss examining the "YAP signaling pathway" but this is not being examined here. In addition, the authors claim that USS restrains YAP in the cytoplasm, but this is not detectable.

As requested by Reviewer 1, we have further conducted qPCR for YAP target genes, the results showed that the YAP staining well represents the “YAP signaling pathway”. We have revised also the “YAP signaling pathway” to “YAP activation”. In terms of detecting whether YAP is in the nucleus or cytoplasm, we have changed the color scheme in Figure 2C. There are clearly green nuclei (solid arrows) and plenty of green in the cytoplasm (dash arrows) outside blue nuclei.

Throughout the study, the authors solely rely on the quantitation of nuclear YAP expression based on IHC. An addition supporting experimental readout is highly recommended, including nuclear/cytoplasmic extraction and western blot to detect YAP in cellular fractions for Figures 3C, and 3D.

We thank the reviewer for the constructive suggestion. As also requested by reviewer 1, we have used the intensity ratios of YAP nuclear/cytoplasmic expressions to quantify YAP activation status for Figure 1, 2, 5. We have also added new Figure 1—figure supplementary 2, which includes qPCR data of YAP target genes THBS1, ANKRD1 and PTX3, as well as YAP upstream in Hippo cascade LATS1 and LATS2.

– For Figures 4E, 4F, 4G, 5E-H, the US control is needed.

We have added US control in Figure 3 (previously 4) and Figure 4 (previously 5).

– In Figure 4B, there is a large decrease in circularity with YAP inhibition, yet proliferation changes appear minimal. What other YAP-dependent mechanisms might contribute to this drastic change in valve morphology? Likewise, Figure 4F indicates 10% pHH3 reactivity, yet this is not represented in Figure 4C. Which cells are proliferating?

Circularity describes the shape, while proliferation changes the size. These are independent parameters that don’t necessarily correlate with each other. We have revised Figure 3 and Figure 4 with higher magnifications to highlight the areas of interest.

– The phenotype in Figure 4D is interesting and unusual. The authors conclude that loss of YAP leads to endothelium folding, but again, there is little evidence to support this. If this is true, then additional supporting readouts are needed including another endothelial cell marker. How do the authors explain an increase in VE-cadherin with so little pHH3 reactivity in the VECs?

We appreciate the reviewer for pointing out this issue. The term “endothelial folding” was poorly chosen. In fact, by “folding”, we meant the shrinkage of surface areas. In fact, by “folding”, we mean the shrinkage of surface areas. Due to the compaction of cushion explants, their surface areas became much smaller, and the shrinkage is approximately proportional to (*d/D*)^2^. Even a small compaction will cause a large reduction in surface areas. As a result, the endothelium at 0-hour was geometrically too large for the cushions at 24-hour. Therefore, the endothelium could cluster into multiple layers Therefore, the increased VE-cadherin expression does not require VEC proliferation.

– On Page 4, the authors comment that VP treatment led to "successful YAP inhibition" but this data is not included.

We thank the reviewer for this reminder. We have included YAP activation data in Figure 3—figure supplementary 1.

– When referring to Figure 5, the authors comment that PY-60 treatment dose-dependently promoted the association of YAP and TEAD. This data is not included.

Other studies have included the dose-dependency of PY-60 induced YAP activation in-vitro. The results showed that a dose of 10 μM increased YAP/TEAD association by 4-fold. Here in our study, we also used 10 μM. The detailed studied of PY-60 is in the reference below.

Shalhout SZ, Yang P-Y, Grzelak EM, Nutsch K, Shao S, Zambaldo C, et al. YAP-dependent proliferation by a small molecule targeting annexin A2. Nature Chemical Biology. 2021;17(7):767-75.

– Figure 5E is confusing and suggests that YAP gain of function has no effect on stress-induced increases in circularity, yet the authors conclude that YAP activation reverses the compaction of valves under TS and U conditions.

We understand the reviewer’s concern, as circularity and compaction may sound correlated. In fact, however, circularity evaluates whether the valves are spherical, while compaction evaluates how small valves became. YAP gain of function maintained a large size, probably due to enhanced cell proliferation, but its shape became round.

– The conclusion of findings from Figure 5D is that "VE-cadherin in the YAP activated endothelium was significantly weaker…showing that YAP activation led to a relaxed endothelium." Again, this conclusion needs to be supported by additional data.

We again appreciate the reviewer for raising this issue. As mentioned in previous answers, the terms “endothelium folding” and “relaxed endothelium” were chosen. Here, the “relaxed endothelium” meant the opposite to the “clustered endothelial geometry”. We have also moved this discussion from the first section of the Results to the Discussion, we have also revised titles of the third and fourth sections of the Results and titles of Figure 3 and Figure 4.

The revised third section of the Results, as well as the title of Figure 3 are as follows:

“Loss of YAP limited cell proliferation and promoted valve shaping

To study the function of YAP in valve growth and remodeling, we added a pharmacological inhibitor of YAP, verteporfin (VP), into the CS (pro-growth) and U conditions. The VP inhibits the interaction between YAP and TEAD, which in turn, blocks transcriptional activation of targets downstream of YAP. (31) We isolated HH34 OFT SL cushion explants and cultured them under CS, CS+VP and U+VP. We confirmed that the concentration of VP we used (5mM) does not influence cell viability (Figure 3—figure supplementary 1A, B). Successful YAP inhibition (Figure 3—figure supplementary 1C) reduced the valve size (Figure 3A vs. Figure 3B), regardless of the media condition. In contrast to the spherical shape of valves cultured under CS, valves cultured in media with VP maintained their trapezoidal shape, which was characterized by circularity (Figure 3E). Further investigation demonstrated that loss of YAP significantly reduces the expression of the proliferation maker, pHH3 in the VIC (Figure 3C, 3F). In addition, the YAP inhibition significantly strengthened the expression of VE-cadherin between VECs (Figure 3D, 3G). Valves normally just had a single layer of endothelium, but the YAP inhibited valves showed five or more layers of endothelium.”

The revised title of fourth section of the Results and Figure 4 is:

“Activation of YAP promoted cell proliferation and inhibited valve elongation”

The new first paragraph of the Discussion is as follows:

“Cardiac valves form in response to mechanical forces generated by the flowing blood. These forces include shear and hydrostatic stress. Our results reveal that they can regulate YAP activity in valvular cells, and the mechanically regulated YAP activity can affect the size, shape and stiffness of valves. First, the shear stress regulates YAP activity in the VEC: OSS (oscillatory shear stress) promotes YAP nuclear translocation while USS (unidirectional shear stress) restricts YAP in cytoplasm. The hydrostatic stress regulates YAP activation in the VIC: CS (compressive stress) activated YAP while TS (tensile stress) deactivated YAP. Secondly, these mechanoresponsive YAP activities have morphological functions. YAP activation in VICs promotes their proliferation and increases valve size, YAP deactivation in VICs leads to valve compaction. In terms of shape, cushion explants tend to form a sphere in a way similar to the clustering of cells to minimize the surface tension (32). By contrast, YAP inhibition promotes valve shaping and leaves a densely packed endothelium. The densely packed endothelium could be a result of the large shrinkage in surface area, during which the endothelium at 0-hour could be geometrically too large for the valves at 24-hour. Consequently, the endothelium could be forced to cluster into multiple layers. Another possibility is that YAP activity can affect cell-cell junctions between VECs. For example, studies reported that YAP can break cell-cell junctions via SMADs mediated TGF-β signaling during the EMT process (33).”

– Figure 5H is mislabeled in the text.

We thank the reviewer for pointing out this mistake, we have corrected the text.

– The authors should comment further on how and why YAP inhibition decreases the strain energy density. How can YAP influence the material properties of avian valves?

We hypothesize that YAP inhibition promotes pro-maturation programs, which increase valve stiffness. We have now added a discussion in the new fourth paragraph of the Discussion:

“In general, YAP works like a mechanobiological switch, converting mechanical signaling into the decision between growth and maturation. When YAP is activated the growth programs are turned on and the maturation programs are suppressed. When YAP is inhibited the growth programs are paused and the maturation programs are released. Unlike the pro-growth function, the pro-maturation side of YAP has been less studied. During valve remodeling, cushions elongate into mature leaflets with increased stiffness. Although both YAP activation and inhibition increased valve stiffness, the stiffness of YAP activated valves was only about half of that of YAP inhibited valves, and only YAP inhibited valves exhibited an in-vivo-like stiffness increase. We have previously shown a linear relationship between valve stiffness and valve maturation (32). This suggests that YAP inhibition promotes a more mature phenotype.”